# Optical Coherence Tomography Angiography Is a Useful Tool for Distinguishing Primary Raynaud’s Phenomenon from Systemic Sclerosis and/or Very Early Disease of Systemic Sclerosis

**DOI:** 10.3390/diagnostics13152607

**Published:** 2023-08-05

**Authors:** Adem Erturk, Ozgur Erogul, Murat Kasikci

**Affiliations:** 1Division of Rheumatology, Department of Internal Medicine, Faculty of Medicine, Afyonkarahisar Health Sciences University, Afyonkarahisar 03030, Turkey; 2Department of Ophthalmology, Faculty of Medicine, Afyonkarahisar Health Sciences University, Afyonkarahisar 03030, Turkey; ozgur.erogul@afsu.edu.tr; 3Department of Ophthalmology, Mugla Training and Research Hospital, Mugla 48000, Turkey; muratkasikci@mu.edu.tr

**Keywords:** systemic sclerosis, Raynaud’s phenomenon, optical coherence tomography angiography, ocular microvasculature

## Abstract

This cross-sectional study aimed to compare optical coherence tomography angiography (OCT-A) findings in patients with primary Raynaud’s phenomenon (PRP; *n* = 22), very early disease of systemic sclerosis (VEDOSS; *n* = 19), and systemic sclerosis (SSc; 25 patients with limited cutaneous SSc (lcSSc) and 13 patients with diffuse cutaneous SSc (dcSSc)). Whole, parafoveal, and perifoveal superficial capillary plexus (SCP) vessel densities (VDs), deep capillary plexus VDs, and whole, inside, and peripapillary VDs were significantly higher in the PRP group (*p* < 0.001). In the lcSSc group, the FAZ perimeter was significantly higher than that in the VEDOSS group (*p* = 0.017). Retinal nerve fiber layer VDs were significantly lower in the lcSSc group than in the PRP and VEDOSS groups (*p* < 0.001). The whole and peripapillary optic disc VDs of the VEDOSS group were significantly higher than in the lcSSc group (*p* < 0.001). Whole SCP VDs (94.74% sensitivity, 100.00% specificity) and parafoveal SCP VDs (89.47% sensitivity, 100.00% specificity) showed the best performance in distinguishing patients with SSc from those with PRP. OCT-A seems to have potential diagnostic value in differentiating patients with PRP from patients with SSc and VEDOSS, and there is potential value in assessing prognostic roles, since findings from OCT-A images could be early indicators of retinal vascular injury long before overt SSc symptoms develop.

## 1. Introduction

Systemic sclerosis (SSc) is a rare connective tissue disease that mainly targets small vessels and causes immune activation, vasculopathy, fibrosis, and reductions in capillary density, with a prominent impact on the skin [1,2]. Organ involvement in SSc may also include organs such as the heart, lungs, kidneys, gastrointestinal system, and eyes. There are two major subsets of SSc: diffuse cutaneous SSc (dcSSc) and limited cutaneous SSc (lcSSc) [2].

Vasculopathy is an important early step in the pathogenesis of SSc, and most patients initially present with Raynaud’s phenomenon (RP) as the first symptom. RP, classified into the primary (PRP) and secondary forms, is a painful, cold-induced episodic, reversible vasospastic condition of the small arteries, which is usually localized in the fingers and toes, affecting approximately 5% of the general population [3]. It has been shown that there are structural changes in the microvasculature before inflammation and fibrosis become clinically evident in patients who develop SSc. It has been stated that RP, which occurs due to vasculopathy, may be a precursor to a possible development of SSc in the future [4,5]. Based on this clinical picture, the concept of very early disease of systemic sclerosis (VEDOSS), which means a very early picture of SSc, has emerged [6]. In a multicenter web-based Delphi exercise, SSc experts developed the preliminary criteria for VEDOSS in 2011. RP, antinuclear antibody (ANA) positivity, and puffy fingers (PFs) were classified as “red flags” or level 1 indicators of VEDOSS in the study. In addition to the “red flag” criteria, the presence of SSc-specific antibodies, such as centromere or Scl-70, and/or abnormal nailfold video-capillaroscopy (NVC) findings in the level 2 assessment was defined as VEDOSS [6,7].

Many ocular findings have been described in patients with SSc, such as dry eye, eyelid fibrosis, keratoconjunctivitis, anterior uveitis, normal-pressure glaucoma, optic neuropathy, retinopathy, and choroidopathy [8,9,10,11]. The prevalence of retinal vascular disease in SSc is reported to be 34–55%, and mild retinal damage may occur long before systemic (or even skin) manifestations occur [8,12,13]. Therefore, detecting early alterations in ocular microcirculation may enable early diagnosis. In this context, assessment of retinal and choroidal vasculature would be ideal for detecting arteriole and capillary pathology in SSc because retinal tissue has the highest oxygen extraction per blood volume, and the choroidal vessels have the highest blood flow in the human body [14]. The gold standard for retinal microvascular investigation has long been recognized as fluorescent angiography (FA) and indocyanine green angiography, which are both invasive methods [15]. In addition, FA is labor-intensive, time-consuming, and difficult to interpret; furthermore, it does not image the radial peripapillary or deep capillary networks. These limitations have increased the popularity of optical coherence tomography angiography (OCT-A), a non-contrast imaging technique that can non-invasively detect angiography and blood flow characteristics of the macular capillary network with high resolution [16]. OCT-A can be used to examine vascular plexuses, inner retina, outer retina, choriocapillaris, and other related areas [17], and it provides the advantage of having less subjectivity and high interobserver reproducibility [18,19]. Prior studies have also shown that capillary density results obtained via NVC and findings from OCT-A were correlated, indicating agreement between the methods [4]. OCT-A was recently shown to detect vascular involvement in systemic diseases, including hypertension, diabetes, systemic lupus erythematosus, and SSc [12,20,21,22,23].

We postulated that OCT-A findings differ among PRP, VEDOSS, and SSc (lcSSc and dcSSc), and these differences can be used to predict patients at high risk of developing overt SSc. The number of studies investigating OCT-A findings in SSc patients is respectable [1,4,12,24,25]; however, to the best of our knowledge, there are only two studies [1,4] comparing OCT-A findings according to SSc subsets. Furthermore, there are no studies that have compared results in patients with PRP, VEDOSS, and SSc. The present study was designed to examine and compare OCT-A findings in patients with PRP, VEDOSS, and SSc (lcSSc and dcSSc) and to determine whether these findings had a diagnostic role.

## 2. Materials and Methods

### 2.1. Study Design, Setting, and Ethics

This cross-sectional study was conducted in the Department of Afyonkarahisar Health Sciences University Faculty of Medicine between October 2022 and February 2023. Ethical approval was acquired from the local ethics committee (decision date: 2 September 2022, decision no: 2022/435) and it was carried out according to the ethical standards stated in the Declaration of Helsinki and its amendments. The method and purpose of the study were explained to all participants in detail, and written informed consent was obtained from each participant.

### 2.2. Study Population

A total of 79 ophthalmological asymptomatic subjects were included in the study, including 22 patients with PRP, 19 patients with VEDOSS, and 38 patients with SSc (25 patients with lcSSc, 13 patients with dcSSc). Patients with rheumatological disease except SSc, patients with glaucoma, cataracts, retinopathy, keratopathy, congenital ocular anomalies, nystagmus, poor eye fixation, and significant media opacity, those who had undergone ocular surgery or laser photocoagulation within the prior 6 months, subjects with systemic vascular diseases (diabetes, systemic hypertension, cardiovascular disease, etc.), pregnant or breastfeeding women, smokers, those in which high-quality OCT-A images had not been obtained due to inadequate pupillary dilation and fixation, individuals with hypersensitivity or intolerance to topical anesthetics or mydriatics, and patients with missing data were excluded from the study. Moreover, patients with other findings that reduced image quality in OCT-A measurements and those who had missing data were not included in the study.

### 2.3. Data Collection, Measurements, and Tools

Age, sex, and other clinicodemographic data were recorded. In addition, ANA patterns, extractable nuclear antigen (ENA) patterns, presence of lung involvement, digital ulcer or pulmonary hypertension, and OCT-A results were recorded.

#### 2.3.1. First Evaluation and Diagnostic Approach

All participants underwent a complete ophthalmic examination that included the best corrected visual acuity, measurement of intraocular pressure with Goldmann applanation tonometry, slit lamp examination, and fundus examination to identify the exclusion criteria. B-scan ultrasonography was performed to evaluate the ocular and orbital structure. Central corneal thickness and axial length were recorded by using a Lenstar LS900 device (Haag-Streit AG, Köniz, Switzerland). The cup-to-disc ratio was calculated as previously described [26]. Cases with pathologies as determined by ocular examination were excluded from the study.

Systemic sclerosis diagnosis was made according to diagnostic criteria of the American College of Rheumatology/European League against Rheumatism (ACR/EULAR) for SSc [27]. A maximum of 34 points can be obtained in this classification system, which has 8 categories. The total score is determined by adding the maximum weight (score) in each category. Patients with a total score of ≥9 were defined as having definitive SSc [27]. The distinction between lcSSc and dcSSc was made in accordance with previously described criteria [28,29]. RP was defined as self-reported or reported by a physician, with a 2-phase color change in fingers or toes consisting of pallor, cyanosis, and/or reactive hyperemia in response to cold exposure or emotion [27]. RP was defined as primary if these symptoms occurred alone without evidence of any associated disorder [3]. Patients with RP, puffy fingers, and ANA positivity were defined as VEDOSS in the presence of SSc-specific autoantibodies such as centromere or Scl-70 and/or abnormal NVC findings (Figure 1) [6,7].

Pulmonary arterial hypertension was diagnosed by right-sided heart catheterization according to standard definitions [27,30]. Interstitial lung disease was diagnosed with the presence of pulmonary fibrosis seen on high-resolution computed tomography or chest radiography when it was most pronounced in the basilar portions of the lungs or according to the occurrence of “Velcro” crackles on auscultation, given that they were not due to another cause such as congestive heart failure [27]. Fingertip ulcer and/or digital pitting scar histories were questioned. Fingertip ulcers were defined as ulcers or scars distal to (or at) the proximal interphalangeal joint, without trauma history. Digital pitting scars were defined as depressed areas in the fingertips as a result of ischemia, not trauma or exogenous causes [27].

#### 2.3.2. Laboratory Analysis

All laboratory measurements were made in the Clinical Microbiology Department of Afyonkarahisar Health Sciences University Faculty of Medicine Hospital in a certified laboratory by using calibrated standard measuring devices. Serum samples were analyzed for ANA and ENA with indirect immunofluorescence by using a substrate kit (EUROIMMUN; Lübeck, Germany) with anti-immunoglobulin G (IgG) antibodies (host: goat, reacts: human). Indirect immunofluorescence patterns were read in a Zeiss Axioscope microscope (Carl Zeiss; Jena, Germany) at 1:40 and 1:100 serum dilutions for ANA positivity. Titers of at least 1:100 were considered positive, and fluorescence patterns were recorded. Sera were tested at 1:40 and 1:100 dilutions for the human IgG class against 15 lines of extremely purified ENAs by using an immunoblot technique according to the manufacturer’s instructions (ANA-Profile 3; EUROIMMUN). ENA results were acquired via the EUROlineScan software (EUROIMMUN) [26].

#### 2.3.3. OCT-A Imaging and Measurements

Images were captured by using the Optovue AngioVue^®^ system (Optovue Inc., Fremont, CA, USA) with an optic-nerve-centered 4.5 × 4.5 mm² field. The images were analyzed by using the AngioAnalytics 2.0 software, which employed the Split-Spectrum Amplitude Decorrelation Angiography algorithm. The imaging parameters included an 840 nm wavelength, 70 kHz scanning frequency, scanning depth of 2–3 mm, and 304 × 304 A-scans with two repeated B-scans at the same location (the lateral discrimination was 15 μm and the axial discrimination was 5 μm). Motion correction and DualTrac were applied throughout the process. The HD Angio Disc (4.5 mm mode) was used to scan an area of approximately 4.5 × 4.5 mm around the optic nerve, while the macula was imaged by using a 3 × 3 mm OCT-A scan. The optical disc 200 × 200 mode was employed to obtain results for the retinal nerve fiber layer (RNFL). The following variables were obtained and recorded for each patient by using the aforementioned protocol: superficial capillary plexus (SCP) vessel densities (whole, fovea, parafovea, and perifovea), deep capillary plexus (DCP) vessel densities (whole, fovea, parafovea, and perifovea), optic disc vessel densities (RNFL global, whole, inside, and peripapillary), foveal avascular zone (FAZ) parameters (total FAZ area, FAZ perimeter, foveal vessel density in a 300-µm-wide region around the FAZ (FD-300)), and flow parameters (outer retinal flow area at a radius of 1–3 mm and the choriocapillary flow area at a radius of 1–3 mm).

The SCP was segmented with an inner boundary of 3 μm below the internal limiting membrane and an outer boundary set at 15 μm below the inner plexiform layer. The DCP was segmented with an inner boundary 15 μm below the inner plexiform layer and an outer boundary at 70 μm below the inner plexiform layer. Choriocapillaris segmentation was defined from 10 μm above to 30 μm below Bruch’s membrane [4]. The VD was defined as the linear length of the vessels divided by the selected area. Perfusion density represents the area of vessel distribution divided by the selected area. The foveal zone was defined as a region with a diameter of 1 mm. The parafoveal zone was defined as a region with a diameter of 3 mm. The perifoveal zone was defined as a region with a diameter of 6 mm [1]. Choroidal thickness measurements were made in the upper, lower, nasal, and temporal regions with selected locations that include those 500 µm, 1000 µm, 1500 µm, and 2000 µm from the fovea. Only high-quality images with a signal strength above 8 were included for analysis. Although both eyes were suitable for our study, only the data of the right eye were evaluated in the final data analysis. All measurements were performed under the same conditions at the same location by the same ophthalmologist (Ö.E).

### 2.4. Statistical Analysis

Statistical analyses were performed by using IBM SPSS Statistics for Windows, Version 25.0 (IBM Corp., Armonk, NY, USA). The normality of continuous variables was assessed by using the Shapiro–Wilk test. Normally distributed variables are presented as the mean ± standard deviation, while non-normally distributed variables are reported as the median (1st quartile–3rd quartile). One-way analysis of variance (ANOVA) and the Kruskal–Wallis test were used for normally and non-normally distributed continuous variables, respectively. Categorical variables were analyzed by using the chi-square test or Fisher–Freeman–Halton test. Post-hoc analyses were adjusted by using Bonferroni correction. Receiver operating characteristic (ROC) curve analysis was used to assess the predictive performance of variables for VEDOSS and SSc, with optimal cut-off points determined by using the Youden index. Statistical significance was set at *p* < 0.05.

## 3. Results

The mean age of the lcSSc group was significantly higher than that of the PRP group (*p* = 0.007), while the remaining groups had similar ages. In addition, there were no significant differences between the groups in terms of sex distribution (*p* = 0.923) (Table 1).

Whole, parafoveal, and perifoveal SCP VDs and DCP VDs were significantly higher in the PRP group than in the other groups (*p* < 0.001 for all). Foveal SCP VDs were significantly lower in the lcSSc group than in the PRP and VEDOSS groups (*p* < 0.001). Foveal SCP VDs were significantly lower in the dcSSc group than in the PRP group (*p* < 0.001). Perifoveal SCP-VDs were significantly higher in the dcSSc group than in the lcSSc group (*p* < 0.001). In the lcSSc group, the FAZ perimeter was significantly higher than that in the VEDOSS group (*p* = 0.017). There were no significant differences between the groups in terms of the outer retina (*p* = 0.758) and choriocapillaris (*p* = 0.121) flow area. RNFL VDs were significantly lower in the lcSSc group than in the PRP and VEDOSS groups (*p* < 0.001). RNFL VDs were significantly lower in the dcSSc group than in the PRP group (*p* < 0.001). The whole, inside, and peripapillary optic disc VDs of the PRP group were significantly higher than those in the other three groups (*p* < 0.001). The whole and peripapillary optic disc VDs of the VEDOSS group were significantly higher than those of the lcSSc group (*p* < 0.001) (Table 2, Figure 2).

The OCT-A measurements that were performed to distinguish SSc patients from PRP patients are presented in Table 3. Whole, fovea, parafoveal, and perifoveal SCP VDs and DCP VDs, RNFL, and whole, inside, and peripapillary optic disc VDs were all able to distinguish patients with SSc from patients with PRP (*p* < 0.001 for all). The whole SCP VDs (94.74% sensitivity and 100.00% specificity) and parafoveal SCP VDs (89.47% sensitivity and 100.00% specificity) showed the best performance.

## 4. Discussion

The main findings of the present study revealed significant differences between various patient groups. Patients with SSc and VEDOSS exhibited notably lower median values of SCP, DCP, and optic disc VDs compared to patients with PRP. Furthermore, within the SSc patient population, those with lcSSc displayed distinct differences in the median values of foveal SCP VDs, RNFL-VDs, whole and peripapillary optic disc VDs, and FAZ perimeter when compared to patients with VEDOSS. Additionally, all SCP, DCP, and optic disc VDs demonstrated significant discriminatory ability when distinguishing patients with SSc and VEDOSS from those with PRP, with the most robust discriminatory performance being observed in whole and parafoveal SCP VDs.

Although its pathogenesis is not fully understood, it is accepted that SSc begins in the microcirculation network [31]. The early detection of ocular microcirculatory involvement in SSc prior to the manifestation of systemic symptoms holds the potential for early diagnosis and implementation of preventive treatments for SSc. This approach can contribute to timely intervention and better management of the disease [1]. Substantial evidence exists for subclinical retinal and choroidal vessel involvement in early-stage SSc [20,32,33]. However, the number of studies investigating OCT-A has remained relatively limited in patients with SSc [1,4,12,24,25]. In the ROC analysis of the current study, all VDs in the SCP, DCP, and optic disc could significantly distinguish patients with SSc and VEDOSS from patients with RP. Whole SCP VDs showed the highest sensitivity and specificity, followed by parafoveal SCP VDs. In another similar study, it was shown that whole, foveal, parafoveal, and perifoveal VDs at the SCP and foveal VDs at the DCP were significantly reduced in the eyes of patients with SSc when compared to healthy controls [1]. As a limitation of the mentioned study [1], only patients with severe systemic hypertension were excluded. However, it is known that hypertension may affect the eye vessels at any stage [34]. In a prospective cross-sectional study, SSc patients without systemic hypertension exhibited significantly lower values of SCP and DCP VDs (whole, fovea, and parafoveal), FAZ areas, FAZ perimeter, foveal density, and 3 mm flow areas in the choriocapillaris compared to the healthy control group. Conversely, the outer retina demonstrated significantly higher 1 mm and 3 mm flow areas [12]. In the study of Mihailović et al., a patient group with VEDOSS, dcSSc, and lcSSc was examined relative to controls. They found that the VDs in the SCP, the choriocapillaris, and the optic nerve head capillary density (inside disc) of the patient group were significantly lower than in the control group, whereas the VDs in the DCP and FAZ did not differ between the groups [4]. Patients with hypertension were also included in the patient group [4]. In another study, the whole and perifoveal DCP VDs of patients with SSc were found to be significantly lower than those of the control group [24]. Rothe et al. showed that, based on OCT-A images, patients with SSc had significantly lower retinal perfusion than that of healthy subjects, and these differences were mainly found in the SCP rather than the DCP [25].

Taking these results together, it can be inferred that retinal and choroidal VDs decrease in patients with SSc, although the results vary across subregions. The findings regarding the FAZ and flow area appear inconsistent. This could be attributed to limitations in OCT-A assessment, as it only captures changes in the choriocapillaris layer and may not detect choroidal changes within the 10 μm range until the onset of retinopathy [12]. The choroidal tissue that can be most clearly visualized with OCT-A is the subfoveal area of the choriocapillaris. However, data from this area are somewhat limited when making assumptions about total choroidal perfusion [35]. Patients with retinopathy were generally excluded from studies. In addition, Carnevali et al. argued that the microcirculation of the choroid is not primarily affected in SSc and may represent a region where obligatory compensatory mechanisms occur to supply the high oxygen demand of the retinal layers [24]. Therefore, the flow area does not seem to be a suitable parameter for evaluating early ocular involvement in SSc. For FAZ-related data, more comprehensive studies are needed.

SSc subsets, PRP, and VEDOSS exhibit distinct clinical and biochemical characteristics. Notably, lcSSc and dcSSc can be differentiated based on the disease progression rate and varying degrees of skin and systemic involvement. These features play a crucial role in distinguishing between the different subsets of SSc [1,28,36]. It can also be expected that the eye involvement will be different between patients with SSc subtypes and those with non-overt SSc [11]. The number of studies examining the differences between RP and SSc, as well as between VEDOSS and definitive SSc, with regard to OCT-A image changes is limited [1,4]. While patients with lcSSc had significantly lower median SCP foveal VDs and optic disc whole, peripapillary, and RNFL VDs compared to patients with VEDOSS, the FAZ perimeter was significantly higher. The median SCP perifoveal VD of the dcSSc group was significantly higher than that of the lcSSc group, and no significant differences were found between the dcSSc group and the VEDOSS group. The reasons for these results were thought to be due to the small number of participants in the dcSSc group or the systemic organ involvement in the dcSSc group. Hekimsoy et al. found no significant difference in OCT-A measures, including SCP and DCP VDs, FAZ, and flow areas, between lcSSc and dcSSc [1]. Mihailović et al. demonstrated that the VD in the choriocapillaris of patients with VEDOSS was significantly lower than that in healthy controls, and the VD of the optic nerve head capillary density in patients with definite SSc was significantly lower than that in VEDOSS patients [4]. Existing data in the literature, including the present study, do not show any difference between lcSSc and dcSSc in terms of OCT-A findings. However, this study suggests that there may be signs of impaired eye involvement in lcSSc when comparing OCT-A data between lcSSc and VEDOSS. However, these results need to be supported by more comprehensive studies.

RP is a key marker that identifies patients at high risk for SSc or other connective tissue diseases and systemic vascular diseases, such as systemic lupus erythematosus (SLE) [5,37]. SLE is more common than SSc and can cause more severe ophthalmic outcomes, such as Purtscher-like retinopathy [38]. Previous studies have also proven that SLE makes significant changes in OCT-A findings [39,40]. Although RP is associated with both diseases, it has been reported that the microvascular consequences of SLE and SSc may be the results of different pathophysiological mechanisms [41]. Patients with RP who exhibit VEDOSS criteria, such as puffy fingers, positive ANA, and scleroderma patterns in NVC, are at a greater risk of developing SSc than those with RP alone [42,43]. However, NVC and FA have the aforementioned disadvantages, and in contrast to these problems, OCT-A has been established to have advantages in terms of interobserver reproducibility, celerity, and ease of use in clinical practice. In this study, we aimed to draw attention to the differences in OCT-A findings between SSc and SSc precursors. The study has the following strengths compared to all or most previous studies on this topic. Systemic diseases such as hypertension and ocular disorders such as retinopathy that could affect OCT-A results were considered as exclusion criteria, creating a subset of patients that could be examined reliably. Patients with PRP, VEDOSS, and SSc were evaluated separately, and the differences in OCT-A data between the groups were examined. The results of the current study showed that OCT-A data, especially SCP, DCP, and optic disc VDs, have the potential to discriminate between PRP and SSc or VEDOSS patients. However, these results should always be carefully considered, and confirmatory studies on a larger study population may be required.

However, it should be noted that there are some limitations to this study. This is a study from a single institution, limiting its generalizability. In the same context, the homogeneity of the study group is another factor that limits the application of these findings in patient groups with characteristics that were defined as exclusion criteria in the present study. In addition, it is a cross-sectional study. Therefore, we cannot arrive at definite conclusions regarding the utility of these results in the assessment of disease progression, which is a result that can only be drawn via prospective studies or from large groups with reliable longitudinal data. The number of subjects is small, but this is largely due to the rarity of SSc, the exclusion criteria, and the single-center design. Despite these limiting factors, the number of participants still remained similar to that in previous studies. The mean duration of the disease and SSc treatment details were not assessed. The potential relationships involving data from eye examinations, NVC measurements, FA results, and autoantibody positivity were not investigated. Since PRP is seen at younger ages, the age of our PRP group was younger than that of the lcSSc group. In addition to these limitations, it is important to acknowledge that the interpretation of different signals from the choroidal vasculature in OCT-A analysis remains a topic of debate, highlighting the need for further research in this area.

## 5. Conclusions

SCP, DCP, and optic disc VDs obtained with OCT-A could significantly distinguish patients with SSc and VEDOSS from patients with PRP. VEDOSS, lcSSc, and dcSSc eyes were found to have significantly lower VDs than those with PRP. It was observed that eyes with lcSSc tended to have lower VDs than eyes with VEDOSS did. OCT-A seems to have potential diagnostic and prognostic value in differentiating patients with PRP from patients with SSc, but the latter role requires further research. Nonetheless, findings from OCT-A images may be early indicators of retinal vascular injury long before patients become symptomatic.

## Figures and Tables

**Figure 1 diagnostics-13-02607-f001:**
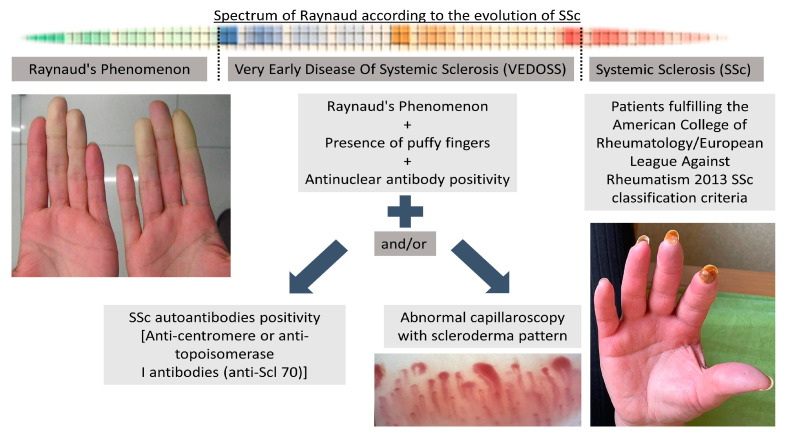
Evolution of SSc from Raynaud’s phenomenon to established SSc.

**Figure 2 diagnostics-13-02607-f002:**
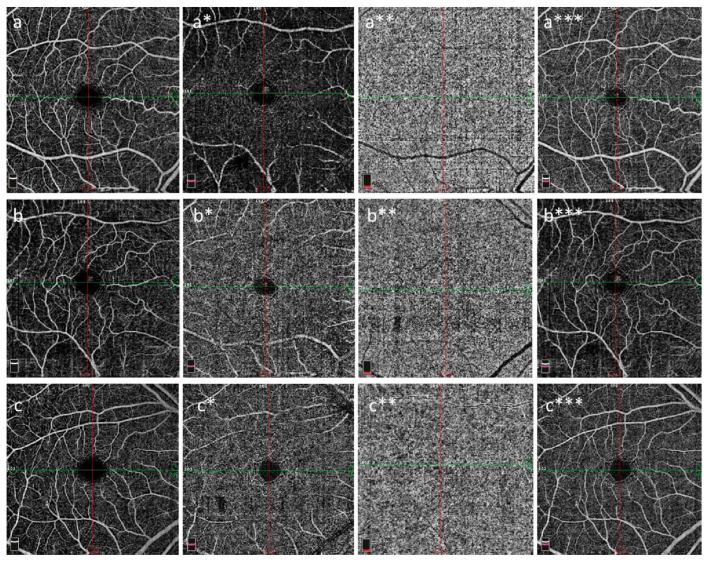
Superficial capillary plexus (**a**–**c**), deep capillary plexus (**a***–**c***), choriocapillaris (**a****–**c****), and FAZ (**a*****–**c*****) in the right eye. OCT-A images from Angio Quickview (6.0 × 6.0 mm scan size and scan quality index = 9/10); PRP (**a**–**a*****), VEDOSS (**b**–**b*****), and SSc (**c**–**c*****) are shown. Note that the SCP and DCP VDs were lower in the cases with VEDOSS and SSc than in the PRP cases. The PRP cases were found to be to the VEDOSS and SSc cases for choriocapillaris (**a****–**c****) and FAZ (**a*****–**c*****). Abbreviations: DCP: deep capillary plexus, FAZ: foveal avascular zone, OCT-A: optical coherence tomography angiography, VEDOSS: very early disease of systemic sclerosis, PRP: primary Raynaud’s phenomenon, SCP: superficial capillary plexus, SSc: systemic sclerosis, VD: vessel density.

**Table 1 diagnostics-13-02607-t001:** Summary of demographics and disease characteristics with regard to diagnosis.

	Diagnosis				*p*
	PRP (*n* = 22)	VEDOSS (*n* = 19)	lcSSc (*n* = 25)	dcSSc (*n* = 13)	
Age	35.95 ± 7.36	42.26 ± 11.06	47.08 ± 14.49 *	42.92 ± 9.81	0.007
Sex					
Female	15 (68.18%)	14 (73.68%)	17 (68.00%)	10 (76.92%)	0.923
Male	7 (31.82%)	5 (26.32%)	8 (32.00%)	3 (23.08%)	
ANA pattern					
Negative	22 (100.00%)	0 (0.00%) *	0 (0.00%) *	0 (0.00%) *	<0.001
Nucleolar	0 (0.00%)	10 (52.63%) *	0 (0.00%) ^#^	12 (92.31%) *^§^	
Centromere	0 (0.00%)	9 (47.37%) *	25 (100.00%) *^#^	0 (0.00%) ^#§^	
Granular	0 (0.00%)	0 (0.00%)	0 (0.00%)	1 (7.69%)	
ENA					
Negative	22 (100.00%)	0 (0.00%) *	0 (0.00%) *	0 (0.00%) *	<0.001
Centromere	0 (0.00%)	9 (47.37%) *	25 (100.00%) *^#^	0 (0.00%) ^#§^	
SCL-70	0 (0.00%)	10 (52.63%) *	0 (0.00%) ^#^	13 (100.00%) *^#§^	
Pulmonary hypertension	0 (0.00%)	0 (0.00%)	6 (24.00%)	2 (15.38%)	0.051
Lung involvement	0 (0.00%)	0 (0.00%)	4 (16.00%)	6 (46.15%) *^#^	<0.001
Digital ulcer	0 (0.00%)	0 (0.00%)	7 (28.00%) *^#^	3 (23.08%)	0.002

Data are given as the mean ± standard deviation or median (first quartile–third quartile) for continuous variables according to the normality of distribution and as the frequency (percentage) for categorical variables. *: significantly different from PRP, ^#^: significantly different from VEDOSS, ^§^: significantly different from lcSSc. Abbreviations; ANA: anti-nuclear antibodies, dcSSc: diffuse cutaneous systemic sclerosis, ENA: extractable nuclear antigen, lcSSc: limited cutaneous systemic sclerosis, PRP: primary Raynaud’s phenomenon, SCL70: anti-topoisomerase 1, VEDOSS: very early disease of systemic sclerosis.

**Table 2 diagnostics-13-02607-t002:** Summary of OCT-A measurements with regard to diagnosis.

	Diagnosis				*p*
	PRP (*n* = 22)	VEDOSS (*n* = 19)	lcSSc (*n* = 25)	dcSSc (*n* = 13)	
SCP-VD, %					
Whole	52.93 ± 2.85	43.19 ± 4.86 *	41.77 ± 2.66 *	41.92 ± 2.46 *	<0.001
Fovea	27.15 (23.5–30.7)	23.8 (19.7–26.2)	12.0 (10.8–17.3) *^#^	16.7 (13.5–23.6) *	<0.001
Parafoveal	54.45 (52.4–55.9)	45.6 (41.2–49.2) *	44.2 (43.7–46.3) *	43.8 (42.4–47.6) *	<0.001
Perifoveal	54.61 ± 3.96	44.83 ± 3.85 *	42.30 ± 1.90 *	46.96 ± 4.20 *^§^	<0.001
DCP-VD, %					
Whole	55.3 (54.1–57.8)	45.5 (40.8–48.7) *	41.2 (40.4–43.8) *	42.5 (41.4–45.9) *	<0.001
Fovea	43.9 (40.4–47.7)	39.2 (33.4–43.3)	33.2 (27.3–41.2) *	36.2 (34.6–43.2)	<0.001
Parafoveal	57.5 (55.5–59.0)	43.1 (41.2–50.4) *	42.5 (40.9–48.6) *	48.3 (45.3–51.5) *	<0.001
Perifoveal	56.95 (54.6–60.1)	42.5 (39.6–50.9) *	41.0 (39.8–45.0) *	41.5 (40.7–48.2) *	<0.001
Foveal avascular zone					
Area, mm^2^	0.29 ± 0.13	0.27 ± 0.10	0.34 ± 0.11	0.27 ± 0.10	0.198
Perimeter, mm	1.94 (1.69–2.35)	2.00 (1.54–2.22)	2.44 (2.13–2.67) ^#^	1.89 (1.77–2.36)	0.017
FD-300 (%)	55.28 (53.55–56.88)	55.88 (51.99–59.80)	51.43 (47.43–58.20)	55.30 (50.40–60.87)	0.449
Flow area, mm^2^					
Outer retina	7.87 (7.65–9.53)	7.57 (6.90–9.40)	7.43 (6.93–9.66)	7.57 (7.31–10.40)	0.758
Choriocapillaris	19.86 (18.82–20.49)	18.99 (17.65–20.11)	19.01 (18.35–19.56)	19.43 (18.44–19.67)	0.121
Optic disc-VD, %					
RNFL global	122.5 (109–128)	109 (101–121)	96 (91–103) *^#^	98 (95–102) *	<0.001
Whole	49.75 (48.9–51.26)	45.4 (41.7–48.8) *	41.2 (40.1–43.2) *^#^	41.2 (39.5–45.4) *	<0.001
Inside	54.8 (52.4–57.2)	42.1 (40.1–48.6) *	34.3 (32.1–46.3) *	44.2 (39.2–47.2) *	<0.001
Peripapillary	54.7 (53.1–57.1)	49.1 (43.0–51.4) *	41.9 (39.8–43.4) *^#^	43.7 (40.4–47.3) *	<0.001
Cup-to-disc ratio	0.11 (0.06–0.19)	0.13 (0.12–0.21)	0.13 (0.11–0.17)	0.07 (0.01–0.14)	0.383

Data are given as the mean ± standard deviation or median (first quartile–third quartile) for continuous variables according to the normality of distribution and as the frequency (percentage) for categorical variables. *: significantly different from PRP, ^#^: significantly different from VEDOSS, ^§^: significantly different from lcSSc. Abbreviations: DCP: deep capillary plexus, dcSSc: diffuse cutaneous systemic sclerosis, lcSSc: limited cutaneous systemic sclerosis, OCT-A: optical coherence tomography angiography, RNFL: retinal nerve fiber layer, PRP: primary Raynaud’s phenomenon, SCP: superficial capillary plexus, VD: vessel density, VEDOSS: very early disease of systemic sclerosis.

**Table 3 diagnostics-13-02607-t003:** Performance of the variables in predicting VEDOSS and systemic sclerosis.

	Cut-Off	Sensitivity	Specificity	Accuracy	PPV	NPV	AUC (95% CI)	*p*
SCP VD, %								
Whole	≤48.4	94.74%	100.00%	96.20%	100.00%	88.00%	0.989 (0.973–1.000)	<0.001
Fovea	≤18.0	54.39%	100.00%	67.09%	100.00%	45.83%	0.818 (0.725–0.910)	<0.001
Parafoveal	≤49.5	89.47%	100.00%	92.41%	100.00%	78.57%	0.962 (0.924–1.000)	<0.001
Perifoveal	≤49.3	91.23%	90.91%	91.14%	96.30%	80.00%	0.954 (0.907–1.000)	<0.001
DCP VD, %								
Whole	≤50.7	92.98%	90.91%	92.41%	96.36%	83.33%	0.916 (0.835–0.997)	<0.001
Fovea	≤38.7	57.89%	90.91%	67.09%	94.29%	45.45%	0.796 (0.682–0.909)	<0.001
Parafoveal	≤52.3	91.23%	90.91%	91.14%	96.30%	80.00%	0.907 (0.810–1.000)	<0.001
Perifoveal	≤53.6	94.74%	86.36%	92.41%	94.74%	86.36%	0.937 (0.877–0.996)	<0.001
Optic disc VD, %								
Retinal nerve fiber layer	≤106	70.18%	90.91%	75.95%	95.24%	54.05%	0.856 (0.771–0.940)	<0.001
Whole VD	≤47.7	85.96%	100.00%	89.87%	100.00%	73.33%	0.935 (0.882–0.988)	<0.001
Inside VD	≤50.5	89.47%	86.36%	88.61%	94.44%	76.00%	0.933 (0.879–0.988)	<0.001
Peripapillary VD	≤51.5	89.47%	95.45%	91.14%	98.08%	77.78%	0.965 (0.929–1.000)	<0.001

Abbreviations; AUC: area under the ROC curve, CI: confidence interval, DCP: deep capillary plexus, NPV: negative predictive value, PPV: positive predictive value, SCP: superficial capillary plexus, VD: vessel density, VEDOSS: very early disease of systemic sclerosis.

## Data Availability

Detailed data are available upon request from the corresponding author.

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
