# Peer review of "Optical Coherence Tomography Angiography Is a Useful Tool for Distinguishing Primary Raynaud’s Phenomenon from Systemic Sclerosis and/or Very Early Disease of Systemic Sclerosis"

_diagnostics, 2023, doi:10.3390/diagnostics13152607_

Round 1
Reviewer 1 Report
The authors have investigated the difference of OCT-A parameters between patients with primary Raynaud's phenomenon (PRP), VEDOSS patients and SSc patients.
There are some remarks and questions related to the paper:
1. The VEDOSS criteria do not include the presence of arthritis, but contain puffy fingers as mandatory criteria (ANA, RP, and puffy fingers are mandatory, and either SSc specific NVC changes of SSc specific antibodies should be also present to fulfill the VEDOSS criteria). Therefore all patients with the current VEDOSS diagnosis should be revised and those with no puffy fingers (either in case history or current), but only arthritis should be excluded from this cohort. The statistics should be recalculated with this new VEDOSS patient cohort if there is difference in the new patient population.
2. Could you elaborate on the differences of the OCT-A findings in SLE (which has a higher prevalence in the population compared to SSc) compared to the OCT-A findings in SSc? SLE patients can also have RP and microvasculopathy.
3. How do you explain that in your SSc cohort practically all patients had decreased SCP-VD and DCP-VD, although in the literature the prevalence of retinal vascular disease was reported to be somewhere between 34 -55%?
4. Was there any difference in duration of Raynaud's phenomenon at the time of OCT exam between the investigated patient cohorts?
5. Do the OCT-A values correlate with the age at time of examination and duration of Raynaud's phenomenon?
6. How many VEDOSS patients had VD based on the OCT-A?
Author Response
July 20th, 2023
Dear Reviewer
We would like to thank you for your useful comments and suggestion given for the early version of our Manuscript. We have revised the manuscript in accordance with the specific and general requests.
Best regards,

Reviewer 2 Report
The article is well-written with a good illustration.
The topic represents a novelty and, once published, would add value to the literature.
The main limitation is the small size of the studied cohort.
Author Response
July 20th, 2023
Dear Reviewer,
Thank you for your valuable feedback on our article. We are grateful for the time you dedicated to reviewing our work and for your positive assessment. Regarding the small size of the studied cohort, we agree with you. As we mentioned in the limitation part of our study, this is largely due to the rarity of SSc, exclusion criteria, and single-center design.
Best regards,
Reviewer 3 Report
Review of Diagnostics [diagnostics-2493885]
Title: Optical coherence tomography angiography is a useful tool to distinguish primary Raynaud’s phenomenon from systemic sclerosis and/or very early disease of systemic sclerosis
Authors: A. Erturk et al.
This paper presents a clinical OCTA study in ophthalmology, aimed at differentiating the different conditions in development of systemic sclerosis (SS) with several OCTA parameters including superficial capillary plexus density and perimeter of foveal avascular zone (FAZ). The study was conducted on 79 patients, of which eyes were examined using a 840 nm wavelength commercial OCT system to reconstruct the complex retinal vasculatures. The OCT findings statistically revealed clear differences between Raynaud’s phenomenon, very early disease of systemic sclerosis, and systemic sclerosis. In my view, this paper is the first OCTA study to quantitatively diagnose the development of systemic sclerosis in stages. Recruiting of cohort, OCTA imaging, and data analysis are systematic and organized. Accordingly, the OCT finding seems also statistically significant and reliable. Discussion is also good.
Hence, the paper could be accepted as a current form without revision.
The paper is very well written in English.
Author Response
July 20th, 2023
Dear Reviewer,
Thank you for your valuable feedback on our article. We are grateful for the time you dedicated to reviewing our work and for your positive assessment.
Best regards,
Reviewer 4 Report
The authors present a cross-sectional study to assess if parameters extracted from optical coherence tomography angiography (OCT-A) images could distinguish between patients with primary Raynaud's phenomenon (PRP), very early disease of systemic sclerosis (VEDOSS) and systemic sclerosis [SSc]. Statistically significant differences were found for some OCT-A parameters suggesting the potential of this imaging technique in differentiating patients with PRP from patients with SSc and VEDOSS.
The paper is well-written. The methodology is adequate and well-described. In particular, the statistical methods used in the work are correct and suitable for the analyzed data. The results are worth reporting and potentially are clinically relevant.
I have just a single comment. The authors state that only data from the right eye were evaluated in the final data analysis. What justified this decision? The relevance of the study findings would be enhanced if those findings were equivalent between the lateral and the contralateral eye. It would be interesting to verify if the performance of some parameters of Table 3, namely fovea and RNFL parameters, increases when considering just the individuals for which the parameter values are equivalent across both eyes. That is, to use information from both eyes to increase the robustness of the OCT-A parameters to predict VEDOSS & systemic sclerosis. This could be easily tested if images from both eyes are still available.
Round 2
Reviewer 1 Report
My previous questions had been answered by the authors.